# Gender diversity and profit efficiency of microfinance institutions: A Sub-Saharan study

**Tarekegn Tariku Ebissa**[1]☯*, **Arega Seyoum Asfaw**[2]☯

1 Department of Accounting and Finance, Wallaga University, Nekemte, Ethiopia, 2 Department of Accounting and Finance, Jimma University, Jimma, Ethiopia

☯ These authors contributed equally to this work.
* tarekegnt.04@gmail.com

## Abstract

Irrespective of the promising opportunity to improve profit efficiency by at least 73%, microfinance institutions operating in Sub-Saharan Africa are efficient only for 27%, far below the average value. The conclusion is drawn after analyzing the profit efficiency of the microfinance institutions using the stochastic frontier approach applied to data obtained from 128 microfinance institutions operating in 34 Sub-Saharan African countries. The study results suggest the presence of uniform profit efficiency experience across time among microfinance institutions. Microfinance institutions operating in low-income countries and credit union form microfinance are economically more efficient than their counterparts. Furthermore, the profit efficiency of microfinance institutions is significantly affected by total assets, cost per loan, loan per staff, legal status, and the county's income group of microfinance. Notably, the profit efficiency of microfinance institutions is adversely affected by the presence of female borrowers and female loan officers suggesting that gender diversity plays a role in the efficiency of microfinance institutions. Finally, we recommend that the managing body of microfinance work more on improving labor efficiency, earning asset utilization, loan collection efficiency, women's involvement and the hottest technology implementation.

## Introduction

Microfinance institutions (MFIs) are vibrant tools for fighting poverty by reaching low-income people and disadvantaged groups such as women by creating affordable and easily accessible financial services. In addition, MFIs play remarkable roles in economic growth by providing services to rural people, reducing transaction costs, easing loan requirements (collateral), smoothing consumption, ensuring gender equality, and lending to small-scale borrowers [1–4]. The roles are more helpful and fruitful in the SSA region than in other parts of the world since the region is occupied by the least banked households and the highest number of low-income people [5]. The good news is that approximately 917 MFIs are currently operating in the SSA region [6].

Furthermore, MFIs have dual missions to achieve in their operations. In the social mission, MFIs are designed to provide cheaper financial services to low-income people, women, and

database via the following link: https://databank. worldbank.org/source/mix-market#.

**Funding:** The author(s) received no specific funding for this work.

**Competing interests:** The authors have declared that no competing interests exist.

those excluded from mainstream financial institutions. In the economic mission, they are expected to generate profit from their operations to maintain a sustainable and stable financial position [2,7]. However, the efficiency of MFIs is a prerequisite to achieving these missions [8]. Improving efficiency by working on profit maximization is an expressway for MFIs to reach their goals. Hence, scholars working inside and outside Africa are urging MFIs operating on the continent to focus on improving their technical efficiency [9], labor productivity [8,10], wise utilization of resources and increasing output [11].

In addition, [11] concluded that MFIs in Sub-Saharan Africa (SSA) are unable to reach people in need of financial services due to inefficiency and multidimensional challenges encountered them in operations. According to [12], MFIs operating in Ghana are observing high regulation costs that adversely affect the outreach of the institutions, and more than half of the MFIs in Africa show a reduction in productivity [2] due to high financial leverage, underinvestment, high financing and labor costs [10]. A study focused on transparency matters of MFIs in SSA [13] highlights the existence of low and highly variable transparency among financial institutions.

Despite the above facts, very few studies have been conducted to investigate the profit efficiency of MFIs operating in SSA, and the empirical gaps are remarkable. Empirical studies in the literature are limited in applying stochastic frontier analysis (SFA)–the parametric approach–[2,11,14], covering a large number of MFIs [4] and assessing the roles of gender diversity in the profit efficiency of MFIs. For instance, a recent study assessed the effect of women's participation on the cost efficiency of MFIs in Sub-Sahara Africa fails to address the profit efficiency aspects of the institutions [15]. Unlike our study, the empirical works of [16–19] focused on addressing the effects of gender diversity on financial performance, social performance and technical efficiency of MFIs in Sub-Saharan Africa. Rather our study emphasized the effects of gender diversity on the profit efficiency of the institutions.

This study contributes to the existing stock of knowledge in three main ways. First, it investigates the profit efficiency of MFIs in SSA from the gender diversity perspectives of borrowing services, board members, management members, loan officers and normal personnel. In this study, the effect of gender diversity on the profit efficiency of MFIs is analyzed from five dimensions of women's engagements. Second, it enhances the generalizability of the study findings by considering a large number of MFIs operating in all SSA countries for longer periods (2009–2018). Finally, it applies an advanced econometric model, i.e., stochastic frontier analysis (SFA).

The study results reveal that firm-specific factors and gender diversity determine the profit efficiency of MFIs. Specifically, the profit efficiency of MFIs in SSA is determined by the firm size, female borrowers, female loan officers, cost per loan, loan per staff, legal status, and the county's income group. The institutions are characterized by implementing expensive procedures to produce a loan, using an inefficient labor force and artificial gender diversity in operations.

## Theoretical and empirical literature review

Profit efficiency represents the ability of a firm to generate optimal profit using essential economic resources in an efficient manner. It measures the distance between a given firm and the best-practiced firm in profit maximization by taking the combination of cost and revenue efficiency into consideration [20]. The profit efficiency of MFIs refers to the maximum profit that a microfinance institution could earn from performing its day-to-day operations.

Theories in efficiency are derived from the concept of optimal utilization of potential resources by a firm possessing the resources. Firms are inefficient when they are not in the

optimum position of using internal resources: labor, capital and information [21]. Different theoretical perspectives are presented in the literature to explain factors contributing to the efficiency or inefficiency of firms. In this study, the theoretical review is presented from the viewpoint of profit efficiency and its determinants; in particular, it focuses on the roles of gender diversity in profit efficiency.

Resource dependency theory and the resource-based view are the frequently mentioned perspectives in the literature regarding the capability of entities to align potential resources with boosting financial performance and efficiency. A basic assumption of resource dependency theory is that self-capability in matching external environmental resources with a firm's decisions and actions enables a firm to improve its financial performance and efficiency [22,23]. The perspective of the resource-based view is derived from the concept of the efficiency and effectiveness of firms in creating sustainable competitive advantage by developing internal resources and capabilities within the firm as well as the optimal use of resources [24].

The board of directors is vested with the power to provide critical resources that meaningfully serve its entity [22]. In other words, the board of directors and the staff of the firm carry out the responsibility of ensuring the successful integration of critical resources and building capabilities inside a firm. Consistent with these perspectives, the attributes (gender diversity, age, educational level and work experience) of the board of directors and the staff determine the actions and decisions taken by a firm and have a notable effect on the profit efficiency of the firm [22,24–26].

In addition, agency theory has been applied in recent empirical studies to define the role of gender diversity in firm efficiency. Agency theory reflects the presence of potential conflicts of interest between shareholders (principal) and management (agent) that arise from the separation of ownership and control of a firm's operations. As one part of corporate governance, diversity in the board of directors and employees reduces the agency problem that may encounter many entities. Consistent with agency theory, [27] suggested the presence of a positive relationship between board gender diversity and firm financial performance. Likewise, a study conducted in China confirmed that female directors improve the investment efficiency of private firms by reducing agency problems and creating disciplined management staff [28]. From an African perspective, [22] stated that board gender diversity enhances the social performance of microfinance institutions by improving board monitoring and lowering cost per borrower and operating costs.

The thoughts of the aforementioned theories are reversed when the view of social identity theory is observed in firms. Social identity theory suggests that individuals may use age and gender as attributes to create their personal category (in-group) and other social groups (out-group) with the desire to either share or deny existing facts [29]. The summarized literature in the study of [30] realized that social categorization maximizes the difference between in-group and out-group and that such social arrangements erode group cohesion, smooth communication and cooperation in organizations. The authors predicted the existence of a negative linear relationship between board gender diversity and firm performance, consistent with the perspective of social identity theory.

On the other hand, existing empirical studies argue that most MFIs operating in SSA countries are technically, socially and economically inefficient [4,8,9,11]. Scholars have suggested the presence of internal and external causes for the outcomes. According to [4], institution age, outreach, productivity and cost per borrower play significant roles in determining the efficiency of MFIs. Likewise, [8,31] stated that MFIs with higher operating expenses, costs per loan and female borrowers are less efficient. In contrast, [14] argues that providing financial services to women and disadvantaged people makes MFIs financially more profitable and sustainable. Furthermore, [11] has suggested that portfolio risk, total assets, return on assets, operational self-sufficiency and yield on gross portfolios are significant determinants of the overall efficiency of MFIs.

Similar study findings do exist for MFIs operating beyond SSA. The study results of [32] reveal that older and larger MFIs are financially more efficient than the their counterparties. According to the study findings of [33], the efficiency of MFI varies based on the regulatory environment, operating region and legal status of the institution.

From gender diversity perspectives, several research findings argue that the contributions of women behind every success are tremendous. For instance, [34] confirmed that women's involvement in research and development teams highly promotes innovation efficiency by providing both informational and social benefits. In their study, [26,35] stated that gender diversity and firm performance are positively interrelated. A study that investigated the association between female leadership and financial performance in MFIs confirmed that the presence of female executives in management and the board significantly improves the financial performance of microfinance institutions [36].

Regarding the efficiency of MFIs, [37] has suggested that female loan officers are key players in enhancing the financial efficiency of MFIs, and [25] has confirmed the presence of a positive association between gender diversity and the social efficiency of MFIs. Despite the wider roles of women in MFIs, empirical studies in SSA are limited in investigating the association between women's roles and the profit efficiency of microfinance institutions. According to [6], women in MFIs act as a client in borrowing services and as a servant for engagement on board members, management members, loan officers and normal personnel.

This study aims to assess the association between women's roles and the profit efficiency of MFIs in SSA from five aspects of gender diversity. Moreover, the study findings contribute to the literature in two other ways. First, it enhances the generalizability of the study findings by taking a large number of MFIs operating in all SSA countries for a longer period (2009–2018). Second, it applies a parametric panel data analysis technique, i.e., the stochastic frontier approach (SFA), unlike those studies that adopted nonparametric data analysis approaches in the MFI literature.

## Material and methods

### Sampling and data source

The main aim of this study is to investigate the profit efficiencies of MFIs existing in SSA countries. All MFIs operating in the SSA region and have been presented their annual financial report to the global microfinance information exchange (MIX–market) database–regulated under the World Bank Group–for at least five concurrent fiscal years from 2009–2018 were considered in this study. Those MFIs that did not present annual reports to the MIX market database for at least five consecutive fiscal years were excluded from this study. In line with this, secondary data were extracted from the updated webpage of the MIX market database for 128 MFIs operating in 34 Sub–Saharan Africa (SSA) countries. Consequently, we are able to obtain an unbalanced panel data set for 930 observations.

### Variable definition & measurement

The variables used in this study were grouped as dependent variables (total profit), input price, output value, and firm- and country-specific factors. The detailed definition and measurement for each variable are presented in Table 1 with essential remarks.

### Model specification

To examine the profit efficiencies of MFIs, the stochastic frontier approach (SFA) was followed in this study based on the stochastic model proposed by [41] and used in [20]. The stochastic

**Table 1. Variable definition & measurement.**

| Panel A: Dependent variable | | |
|---|---|---|
| | Measurement | Definition |
| π: total profit | Total income–total cost [20] | Net income before tax & donation = Total revenue less total expenses during a given period, including operating and nonoperating. Tax expense and donation income are not considered in the calculation [6]. |

| Panel B: Independent variables | | |
|---|---|---|
| Input prices | | |
| Labor: Price of labor | Personnel expenses divided by total number of personnel [6]. | Cost paid as a salary or wage to mobilize employees' effort |
| Fund: Price of fund | *Interest expenses (interest paid) divided by total deposits and borrowings [6]. | *Interest expense = interest expense on deposits + interest expense on borrowings |
| PPC: Price of physical capital | *Operating expense divided by net fixed assets** | *Operating expense includes expenses not related to personnel expenses, depreciation, amortization and administrative expenses & **Net fixed assets are tangible assets net of accumulated depreciation [6]. |
| Output quantities | | |
| Loan: Net loan portfolio | Value of loan portfolio net of impairment loss allowance and unearned income and discount (when applicable) [6]. | Describes efficiency of MFI in minimizing adverse outcomes |
| OEA: Other earning Assets | Total of all other assets | Includes receivables, long-term investment, inventories, intangible assets [6]. |

| Panel C: Determinants of inefficiency/efficiency | | |
|---|---|---|
| Firm-specific variables | | |
| lnfirmsize: Firm size | Natural logarithm of total assets [20]. | Total assets [6]. |
| ROA: Profitability | ROA [8,20]. | Return on asset [6] |
| Cost: Cost per loan | Operating expense/avg. number of loan | Productivity and efficiency [6]. |
| Loan: Loan per staff | Number of loan outstanding/number of staff | Productivity and efficiency [6] |
| Boardg: Broad gender diversity | Percent of female as board members [6,38] | Gender diversity in corporate governance |
| Mgmtg: Management gender diversity | Percent of female in management [6,38] | |
| Femaleb: Female borrower | Percent of female as active borrowers [6–8] | Efficiency in outreach (depth of outreach) |

| Panel D: Controlling and additional variables | | |
|---|---|---|
| Time trend in year (T) | Labeled as "1" if 2009, "2" if 2010, "3" if 2011 . . . and "10" if 2018. | Added as dummy variable |
| Income: Income category | 1 = Upper middle income<br>2 = Low middle income<br>3 = Low income | MFI's country income category (dummy variable) [39] |
| Type: Type of MFIs | 1 = NBFI, 2 = Bank, 3 = NGO, 4 = Credit union/ cooperative [40] | Legal status of MFI [6] |

frontier approach is a parametric approach that allows researchers to analyze panel data for stochastic production where the disturbance terms, a mixture of an inefficiency term and the idiosyncratic error, are diagnosed separately. Using SFA rather than nonparametric approaches (such as Data Envelopment Analysis (DEA)) has remarkable advantages because it allows efficient estimation of efficiency levels by separating inefficiency from other stochastic shocks and introducing country- or firm-specific controlling variables into the stochastic frontier model [20]. In proposing a stochastic frontier model, [41] assumed that the output deviates from the optimal frontier line as a result of the natural disturbance error (random shocks) and actual inefficiency in accomplishing activities. Thus, stochastic frontier models allow us to

estimate the efficiency of a particular firm separately by controlling for random shocks and inefficiency levels.

Subsequently, the translog profit function was applied in estimating the profit ($\pi$) efficiency of MFIs because it has the advantage of presenting in more flexible functional form. The translog stochastic profit function was introduced after the following adjustments were made. First, to avoid an invalid outcome of the logarithm function for negative numbers, profit ($\pi$)–dependent variable–is transformed as $\ln(\pi + \theta + 1)$, where $\theta$ equals the minimum profit amount of MFI in absolute value term, and MFI incurred maximum loss (earned minimum profit) from the overall sample will have zero profit after transformation, i.e., $\ln(1) = 0$.

Second, the composite error ($\varepsilon_{ijt}$) equals $v_{ijt} - u_{ijt}$ for the profit function. Third, the profit efficiency ($\pi_{it}$) score is defined by $\pi_{it} = exp(-u_{it})$ with values between zero and one–a value closer to one indicates more profit efficiency [20]. Fourth, the alternative profit function approach was followed in this study in the output quantity selection procedure since the presence of valid data for the output price is very rare in the study area. Finally, a linear homogeneity assumption is imposed on the input prices of labor and funds as well as the total profit by normalizing them in terms of the price of physical capital before taking their logarithms [20,42]. By considering these specifications, the following profit efficiency frontier model was introduced.

$$\pi_{ijt} = f\left(z_{ijt}\right) + \varepsilon_{ijt} \tag{1}$$

$$\varepsilon_{ijt} = v_{ijt} - u_{ijt} \tag{2}$$

In the stochastic frontier model of [41], the profit function of $MFI_i$ operating in specific country j across time period $t$ is defined in terms of the explanatory variables ($z_{ijt}$) and the disturbance term ($\varepsilon_{ijt}$). The disturbance term is further divided into a random shock ($v_{ijt}$) and an actual inefficiency term($u_{ijt}$). Eq 3 presents the translog stochastic profit function, and a detailed description and measurement for each variable used are presented in Table 1.

$$
\begin{aligned}
ln\left(\frac{\pi}{PPC}\right)_{ijt} &= \alpha_0 + \alpha_1 ln\left(\frac{Labor}{PPC}\right)_{ijt} + \alpha_2 ln\left(\frac{Fund}{PPC}\right)_{ijt} + \alpha_3 ln(Loan)_{ijt} + \alpha_4 ln(OEA)_{ijt} \\
&+ \alpha_5\frac{1}{2}\left[ln\left(\frac{Labor}{PPC}\right)\right]^2_{ijt} + \alpha_6\frac{1}{2}\left[ln\left(\frac{Fund}{PPC}\right)\right]^2_{ijt} + \alpha_7\frac{1}{2}[lnLoan]^2_{ijt} + \alpha_8\frac{1}{2}[lnOEA]^2_{ijt} \\
&+ \alpha_9 ln\left(\frac{Labor}{PPC}\right)_{ijt} * ln\left(\frac{Fund}{PPC}\right)_{ijt} + \alpha_{10} ln\left(\frac{Labor}{PPC}\right)_{ijt} * ln(Loan)_{ijt} \\
&+ \alpha_{11} ln\left(\frac{Labor}{PPC}\right)_{ijt} * ln(OEA)_{ijt} + \alpha_{12} ln\left(\frac{Fund}{PPC}\right)_{ijt} * ln(Loan)_{ijt} \\
&+ \alpha_{13} ln\left(\frac{Fund}{PPC}\right)_{ijt} * ln(OEA)_{ijt} + \alpha_{14} ln(Loan)_{ijt} * ln(OEA)_{ijt} \\
&+ \alpha_{15} ln\left(\frac{Labor}{PPC}\right)_{ijt} * T + \alpha_{16} ln\left(\frac{Fund}{PPC}\right)_{ijt} * T + \alpha_{17} ln(Loan)_{ijt} * T \\
&+ \alpha_{18} ln(OEA)_{ijt} * T + \alpha_{19} T + \frac{1}{2}\alpha_{20} T^2 + u_{ijt} \\
&- v_{ijt}
\end{aligned}
\tag{3}
$$

The stochastic frontier approach assumes that total profit deviates from the targeted profit as a result of a random disturbance term $v_{ijt}$ and the inefficiency term $u_{ijt}$ [41].

The $v_{ijt}$ represents a truncated random error due to measurement error from explanatory variables and is assumed to be independent and identically distributed from $u_{ijt}$ with N (0, $\sigma_v^2$). $u_{ijt}$ represents the nonnegative random variable estimates inefficient effect and is assumed to follow an asymmetric half normal distribution in which both the mean $u$ and variance $\sigma_u^2$ are varied. Furthermore, parametrization techniques suggested by [41] and used in [20] for $\sigma_v^2$ and $\sigma_u^2$ are applied in this study; these are $\sigma^2 = \sigma_v^2 + \sigma_u^2$ and $\gamma = \sigma_u^2/(\sigma_v^2 + \sigma_u^2)$.

According to [41], a one-step stochastic frontier model can be used to identify predictors of the efficiency of a firm. The stochastic frontier approach uses the maximum likelihood estimation technique to predict parameters included in the frontier model. In this study, the following alternative model is formulated to assess determinants of profit efficiency of MFIs after estimating efficiency scores through translog stochastic profit function.

$$
\begin{aligned}
u_{ijt} &= \beta_0 + \beta_1 lnfirmsize_{ijt} + \beta_2 lnROA_{ijt} + \beta_3 lncost_{ijt} + \beta_4 lnloan_{ijt} + \beta_5 Boardg_{ijt} \\
&\quad + \beta_6 Mgmtg_{ijt} + \beta_7 \text{ln Femalestaff}_{ijt} + \beta_8 Femaleof_{ijt} + \beta_9 Femalebr_{ijt} \\
&\quad + \beta_{10-12} Type_{ijt} + \beta_{13-14} income_{ijt} + z_{ijt}
\end{aligned}
\tag{4}
$$

where $ln$ is the natural logarithm function, *firmsize* is total assets of MFI, *ROA* is the return on assets, *Boardg* is board gender diversity in %; *Mgmtg* is management gender diversity in %, *femalestaff* is the number of normal female personnel and *Femalebr* is the proportion of women borrowers in %. Cost is the cost incurred per loan; loan is the number of loans produced by each staff member; Type is the legal status of MFI, *income* is the country's income group and $z_{ijt}$ is the disturbance term in the estimation of profit efficiency determinants.

## Results and discussion

### Attributes of profit efficiency of MFIs in SSA

The stochastic frontier approach was employed to assess the attributes and determinants of profit efficiency of MFIs operating in SSA using ten-year penal data from 2009–2018. To maintain the coherence of idea flow, the discussion begins in this section by being classified into two main sections. The first section presents the basic attributes of profit efficiency, and the second section summarizes the factors affecting the profit efficiency of MFIs in SSA countries. Analysis regarding attributes of profit efficiency of MFIs in SSA was made based on evidence presented in Table 2. The overall profit efficiency of MFIs in SSA was 27% on average, and the score reveals the existence of a tremendous vacuum for further improvement in profit efficiency. In other words, MFI operating in SSA has a great opportunity to enhance its profit efficiency by at least 73%.

Even though there was no marked difference in profit efficiency score among other types of MFIs, the credit union form of MFIs was more efficient in generating profit (35%), whereas nonbank microfinance institutions (NBFIs) were less efficient (23.5%) on average value. MFIs operating in lower middle-income (25%) and low-income (29%) countries are more efficient in generating profit than those operating in upper-middle income countries (21%), on average.

Regarding the time trend, the profit efficiency of the institutions runs between 25% and 27% on average. The trend reveals the presence of uniform profit efficiency experience over the study period. In another expression, there is no adequate effort from MFIs dedicated to improving profit efficiency from time to time in line with existing technical and technological advancement, regardless of the existing space for further improvement of at least 73%. These findings are consistent with the study of [4,5] but contradict the findings of [8,40].

**Table 2. Summary statistics for profit efficiency scores.**

| Panel A: Types of MFI | Profit efficiency scores (%) | |
|---|---|---|
| | Mean | Std.Dev. |
| NBFI | 23.50 | 0.077 |
| Bank | 26.33 | 0.120 |
| NGO | 25.90 | 0.129 |
| Credit union/cooperative | 34.90 | 0.131 |
| Overall efficiency | 27.00 | 0.122 |
| Panel B: Income category of MFI's country | | |
| Upper-middle income | 21.45 | 0.093 |
| Lower-middle income | 25.44 | 0.119 |
| Low income | 29.29 | 0.123 |
| Overall efficiency | 27.00 | 0.122 |
| Panel C: Time trend (2009–2018) | | |
| 2009 | 27.43 | 0.139 |
| 2010 | 27.13 | 0.135 |
| 2011 | 27.58 | 0.135 |
| 2012 | 27.62 | 0.125 |
| 2013 | 27.46 | 0.133 |
| 2014 | 26.70 | 0.122 |
| 2015 | 26.56 | 0.109 |
| 2016 | 26.56 | 0.103 |
| 2017 | 26.20 | 0.104 |
| 2018 | 25.94 | 0.095 |
| Overall efficiency | 27.00 | 0.122 |

Source: Authors' computation.

Furthermore, the frontier model results presented in Table 3 show the presence of meaningful statistical relationships among input prices, output quantity, and the profit efficiency of MFIs. The profit frontier model is statistically significant and acceptable for analysis for three reasons (see Table 3). First, the chi-square test of zero coefficient variation in a model was rejected at a 1% significance level ($x^2 = 718.29$), which implies that the explanatory variables used have significantly explained the existing variations in the model and that the parameter coefficients are significantly different from zero. Second, the value of sigma-squared ($\sigma^2 = 8.504$) was significant at a 1% significance level, implying that the estimate of parameters is highly significant. Third, the estimated values of Gamma ($\gamma = 0.6414$) in the model were also highly significant at a 1% significance level, which implies that a significant amount of variation is derived from the inefficiency of the MFIs, while variance due to random error is small.

The estimated parameter for input prices and output quantities shows the existence of an insignificant linear relationship between labor price, loan, and profit efficiency. However, the effect of input price for funds was positive ($\alpha_3 = 18.792$) and highly significant at a 1% significance level, whereas the effect of other earning assets ($\alpha_4 = -0.244$) was negative and significant at a 5% significance level. These results have important implications regarding interest expense management and the balance sheet (asset) utilization capability of financial institutions. First, effective management of cost paid for a fund in the form of interest expense enhances profit efficiency of the MFIs. Second, a portion of the inefficiency of the MFIs is sourced from underutilization of assets on hands, specifically; other earning assets existing

**Table 3. Stochastic frontier regression results.**

| Dependent variable–Total Profit | | Profit efficiency | |
|---|---|---|---|
| **Panel A: Input, Outputs and Cross terms** | | | |
| **Notation** | **Parameter** | **Coef.** | **t value** |
| ln(Labor/PPC) | $\alpha_1$ | -0.002 | -0.01 |
| ln(Fund/PPC) | $\alpha_2$ | 18.792 | 2.60*** |
| ln(Loan) | $\alpha_3$ | 0.136 | 0.99 |
| ln(OEA) | $\alpha_4$ | -0.244 | -2.11** |
| ½ ln(Labor/PPC)^2 | $\alpha_5$ | 0.137 | 12.94*** |
| ½ ln(Fund/PPC)^2 | $\alpha_6$ | -11.834 | -1.42 |
| ½ ln(Loan)^2 | $\alpha_7$ | -0.005 | -1.11 |
| ½ ln(OEA)^2 | $\alpha_8$ | 0.004 | 0.62 |
| ln(Labor/PPC)* ln(Fund/PPC) | $\alpha_9$ | -2.798 | -7.11*** |
| ln(Labor/PPC)* ln(Loan) | $\alpha_{10}$ | -0.029 | -1.91* |
| ln(Labor/PPC)* ln(OEA) | $\alpha_{11}$ | 0.004 | 0.40 |
| ln(Fund/PPC)* ln(Loan) | $\alpha_{12}$ | -0.216 | -0.41 |
| ln(Fund/PPC)* ln(OEA) | $\alpha_{13}$ | 0.701 | 2.21** |
| ln(Loan)* ln(OEA) | $\alpha_{14}$ | 0.011 | 1.40 |
| ln(Labor/PPC)*T | $\alpha_{15}$ | 0.009 | 1.74* |
| ln(Fund/PPC)*T | $\alpha_{16}$ | 0.407 | 2.27** |
| ln(Loan)*T | $\alpha_{17}$ | -0.006 | -1.21 |
| ln(OEA)*T | $\alpha_{18}$ | -0.003 | -0.73 |
| ½(T)^2 | $\alpha_{19}$ | 0.01 | 2.57** |
| T | $\alpha_{20}$ | -0.027 | -0.40 |
| Constant | $\alpha_0$ | 16.512 | 9.47*** |
| Wald Chi-square | | 718.29 | 0.000*** |
| Sigma squared | | 0.4159 | 8.504*** |
| Gamma (γ) | | 0.6414 | 14.356*** |
| Log-likelihood function | | -611.231 | |
| Number of observation | | 930 | |
| Number of group | | 128 | |

*** p<1%,

** p<5%,

* p<10%.

Source: Authors' stochastic regression outputs.

with the institutions were not producing adequate profit to the level they are expected to produce rather exposing them to other expenses due to nonperforming loans and depreciation.

Moreover, multiplicative input and output terms have both negative (refer to the coefficient of $\alpha_9$ and $\alpha_{10}$) and positive (refer to the coefficient of $\alpha_{13}$) significant impacts on profit efficiency, implying the existence of spaces for further improvement. The effects of labor and fund price become significant over time. The profit efficiency of MFIs in SSA improves over time as institutions' work experience increases in the effective utilization of labor and funds, and there is a quadratic relationship between the time trend and profit efficiency.

## Profit efficiency determinants

In this section, determinants of profit efficiency of MFIs in SSA are discussed based on regression outputs presented in Table 4. Three different empirical modes are developed to

**Table 4. Factor affecting profit efficiency of MFIs in SSA.**

| Dependent variable: Profit Efficiency | | Model 1 | | Model 2 | | Model 3 | |
|---|---|---|---|---|---|---|---|
| Independent variable | Parameter | Coef | t value | Coef | t value | Coef | t value |
| Lnfirmsize | $\beta_1$ | 0.007 | 3.09*** | 0.011 | 5.35*** | | |
| lnROA | $\beta_2$ | 0.012 | 1.38 | 0.008 | 0.92 | | |
| Lncost | $\beta_3$ | -0.033 | -7.32*** | -0.033 | -7.07*** | | |
| Lnloan | $\beta_4$ | -0.029 | -4.83*** | -0.031 | -5.14*** | | |
| Lower_middle_income | $\beta_{10}$ | 0.009 | 0.32 | 0.038 | 1.29 | | |
| Low_income | $\beta_{10}$ | 0.043 | 1.45 | 0.068 | 2.29** | | |
| NBFI | $\beta_{11}$ | -0.109 | -10.15*** | -0.113 | -10.34*** | | |
| Bank | $\beta_{11}$ | -0.071 | -5.75*** | -0.073 | -5.83*** | | |
| NGO | $\beta_{11}$ | -0.066 | -6.01*** | -0.083 | -7.69*** | | |
| Boardg | $\beta_5$ | 0.002 | 0.15 | | | -0.010 | -0.60 |
| Mgmtg | $\beta_6$ | -0.001 | -0.08 | | | 0.002 | 0.13 |
| Femalebr | $\beta_7$ | -0.070 | -4.31*** | | | -0.088 | -5.59*** |
| lnFemalestaff | $\beta_8$ | 0.006 | 2.49** | | | 0.003 | 1.44 |
| Femaleof | $\beta_9$ | -0.073 | -4.20*** | | | -0.050 | -2.71*** |
| Constant | $\beta_0$ | 0.480 | 5.36*** | 0.385 | 4.88*** | 0.326 | 8.08*** |
| | Number of obs. | 930 | | 930 | | 930 | |
| | Chi-square | 274.91*** | | 214.410*** | | 53.079*** | |
| | Log likelihood function | 757.38 | | 733.42 | | 662.76 | |

*** $p<1\%$,

** $p<5\%$,

* $p<10\%$.

Source: Authors' stochastic regression outputs.

understand the impacts of firm-specific factors and gender diversity on the profit efficiency of MFIs. The first model reveals the combined effects of firm-specific factors and gender diversity on the profit efficiency of MFIs. The second model presents the impacts of firm-specific factors, and the third model shows the separate outcomes for gender diversity in relation to profit efficiency. In other words, the determinants are identified after regressing firm size, ROA, board gender diversity, management gender diversity, female borrowers, female staff, cost of the loan, loan per staff, type of MFI, and country-specific factor (income group) on profit efficiency. The results are obtained from the corresponding frontier function through maximum log-likelihood estimation.

The parameters of the chi-square test and log likelihood function confirm the significance of all models in explaining the existing variations, and the coefficients of all parameters are different from zero at a 1% significance level. Moreover, except for the low-income group and female staff, the effects of other explanatory variables remain consistent across all models, confirming the robustness of the study findings. Thus, it is acceptable to use the estimated parameters in explaining determinants of MFI profit efficiency. The explanation is presented in this section.

Regarding firm-specific determinants, the study results reveal the presence of a significant and positive association between total assets (firm size) and profit efficiency. The association is highly significant at the 1% significance level. The profit efficiency of MFIs increases as the total assets of the institution increase. Thus, efficient utilization of economic resources (assets) enhances the profit efficiency of MFIs operating in SSA.

As expected, the effects of cost per loan and loan per staff were negative and statistically significant at a 1% significance level. Cost per loan and loan per staff are deteriorating the profit efficiency of MFIs in SSA. The findings have two remarkable implications. First, microfinance institutions in SSA are exercising the costly practice of producing a loan that harms the profitability of the institutions. Second, staff engaged in producing loans are not generating adequate loans to the expected level. As a result, labor forces running microfinance are not productive on one side and expensive on the other side. In general, MFIs in SSA are producing loans at a cost greater than they will incur, and the existing staff are producing loans below the expected level. This result partially confirms the finding of [8].

Furthermore, the type of MFI and its country income group significantly determine the profit efficiency of MFIs in SSA. The credit union (cooperative) form of MFIs significantly earns more positive profit than other forms of MFIs, and those operating in low-income countries were more efficient in earning profit than those operating in other income groups. In line with the studies of [33,40], the legal status of MFIs and the country's income group do matter for the profit efficiency of the institutions.

In addition to addressing the impacts of the aforementioned factors, this study aims to investigate the effects of gender diversity on the profit efficiency of MFIs in SSA from five perspectives of gender diversity. The effects are addressed in terms of women engagements on the board member, management, borrowing services, loan officers and regular personnel (refer to model 3). The study results reveal that the presence of women on board members, management and normal staff has no significant effect on the profit efficiency of microfinance institutions; however, female borrowers and loan officers significantly affect profit efficiency.

The mission to reach female borrowers in SSA adversely affects the profit efficiency of MFIs. There is a significant and negative association between the presence of female borrowers and the profit efficiency of institutions. Although the evidence is too weak to generalize, the result confirms the presence of trade-off practices between the profit efficiency of MFIs and financial services provision to women. This study's finding is consistent with the finding of [7,31,43].

However, the justification is different. In their study, [11] suggested that ensuring operational efficiency is a prerequisite for MFIs in SSA to simultaneously achieve their dual missions–reaching the disadvantage group and maintaining financial sustainability. Hence, we argue that the presence of adverse effects between female borrowers and profit efficiency is not only because of serving women in financial services provision. As observed in the above discussion, holding a less productive labor force and costly practices of producing loans fundamentally deteriorate the profit efficiency of MFIs. This implies that MFIs in SSA are not efficient in their operation and provision of borrowing services to customers, including women borrowers. In other words, inefficient borrowing services provision has produced inefficient borrowers that harm the profit efficiency of the MFIs.

In contrast to our expectations and previous empirical findings [37,44], female loan officers do not contribute to the profit efficiency of MFIs operating in SSA. The study finding discloses the presence of a negative and significant association between female loan officers and the profit efficiency of MFIs.

Theoretically, these empirical findings present notable remarks. The adverse effects of borrowers' and loan officers' gender diversity against the profit efficiency of MFIs contradict the concepts of resource dependency theory, resource-based view and agency theory. However, the findings are consistent with the concept of social identity theory. It seems that the institutions serve women in borrowing services and allow them to hold officer positions, probably for the desire to meet regulation requirements for gender diversity at work. As a result, women receiving borrowing services from the institutions and working in the institutions may face

group isolation and ignorance by their counterparts, and the institutions are limited in effectively extracting and using the inherent potential of women in the workplace.

Practically, this implies that women's empowerment and capacity building are not part of the financial service provisions for MFIs in SSA. The institutions provide borrowing services to those women who meet the borrowing requirements in the absence of regular follow-up, counseling and training that capacitate women borrowers and help them repay their loan when it dues. Furthermore, MFIs are not active in empowering women officers. Giving an appointment to women to act as a loan officer may not be adequate to extract the inherent potential of women in decision-making, resource provisions and cost management. It requires effective assistants to build self-confidence and take full responsibility.

## Conclusions and managerial implications

MFIs operating in SSA realize profit efficiency below the average, only 27%. It indicates the presence of potential opportunities to raise the current profitability position of MFIs in SSA by improving their efficiency level by 73%. Credit union MFIs and MFIs operating in low-income countries are more efficient in realizing profit efficiency than other forms of MFIs and institutions operating in the upper-middle-income group. A uniform profit efficiency level was recognized by the MFIs in SSA from time to time due to the absence of self-advancement and management in line with the rapid changes in technology, and financial services provision techniques. Input prices such as the labor price and price of a fund and the output item, particularly other earning assets, play significant roles in the profit efficiency of MFIs operating in SSA.

Moreover, firm size, cost per loan, loan per personnel, legal status, and country's income group significantly affect the profit efficiency of MFIs in SSA. Regarding gender diversity, the profit efficiency of microfinance institutions is adversely affected by the presence of female borrowers and female loan officers. It is necessary to work more on improving profit efficiency by designing alternative mechanisms that could enhance labor productivity, reduce loan-related costs, capacitate women borrowers and increase loan collection efficiency.

To improve profit efficiency, MFIs in SAA are expected to undertake the following managerial actions and decisions. First, capacity-building reforms and training designed to enhance labor productivity, ensure effective utilization of assets and reduce loan-operating costs should be implemented. Second, redesign human resources policy and loan provision regulations to ensure women's involvement on the board member, management, personnel and borrowing services enable the MFIs to take advantage of gender diversity in making sound decisions, resource provisions and cost management. Finally, focus on removing work environment practices exposing women to self-categorization and isolation and deteriorating their confidence to take full responsibility in decision-making.

To increase the generalizability of the study findings, future research is recommended to incorporate macroeconomic factors and apply an instrumental variable panel data approach in addition to SFA to rigorously investigate endogeneity problems and assess the consistency of the SFA findings.

## Supporting information

**S1 File.**
(XLSX)

## Author Contributions

**Conceptualization:** Tarekegn Tariku Ebissa, Arega Seyoum Asfaw.

**Data curation:** Tarekegn Tariku Ebissa.

**Formal analysis:** Tarekegn Tariku Ebissa.

**Methodology:** Tarekegn Tariku Ebissa.

**Software:** Tarekegn Tariku Ebissa.

**Supervision:** Arega Seyoum Asfaw.

**Validation:** Tarekegn Tariku Ebissa.

**Visualization:** Arega Seyoum Asfaw.

**Writing – original draft:** Tarekegn Tariku Ebissa.

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
