## [Decision Letter · Decision Letter 0]

6 Feb 2024

PONE-D-23-42063Gender diversity and profit efficiency of microfinance institutions: A Sub-Saharan studyPLOS ONE

Dear Dr. Ebissa,

Thank you for submitting your manuscript to PLOS ONE. After careful consideration, we feel that it has merit but does not fully meet PLOS ONE’s publication criteria as it currently stands. Therefore, we invite you to submit a revised version of the manuscript that addresses the points raised during the review process.

**ACADEMIC EDITOR: ****1. **Major revision is required as per reviewer reports.**2. **As indicated by the reviewer, we request the author to engage with a professional English editor to proofread the article. Please provide us the anonymous certificate of proofreading and attached it in the table of correction.

We look forward to receiving your revised manuscript.

Kind regards,

Jasman Tuyon, Ph.D., MBA

Academic Editor

PLOS ONE

A clean copy of the edited manuscript (uploaded as the new *manuscript* file).

Additional Editor Comments:

Major revision as per the reviewer reports

Reviewers' comments:

Reviewer's Responses to Questions

**Comments to the Author**

1. Is the manuscript technically sound, and do the data support the conclusions?

Reviewer #1: Yes

Reviewer #2: Partly

2. Has the statistical analysis been performed appropriately and rigorously? 

Reviewer #1: Yes

Reviewer #2: Yes

3. Have the authors made all data underlying the findings in their manuscript fully available?

Reviewer #1: No

Reviewer #2: Yes

4. Is the manuscript presented in an intelligible fashion and written in standard English?

Reviewer #1: Yes

Reviewer #2: Yes

5. Review Comments to the Author

Reviewer #1: Title of manuscript: Gender diversity and profit efficiency of microfinance institutions: A Sub-Saharan study

GENERAL COMMENTS

The abstract of the study which I reproduce below sums up the focus of the study:

Irrespective of the promising opportunity to improve profit efficiency by at least 73%,

microfinance institutions operating in Sub-Saharan Africa are efficient only for 27%, far below the average value. The conclusion is drawn after analyzing the profit efficiency of the microfinance institutions using the stochastic frontier approach applied to data obtained from 128 microfinance institutions operating in 34 Sub-Saharan African countries. The study results suggest the existence of uniform profit efficiency experience from time to time due to the lack of sufficient effort exerted to improve efficiency concurrent with the changes in technology and techniques of financial services provision. Microfinance institutions operating in low-income countries and credit union form microfinance are economically more efficient than their counterparts. Furthermore, the profit efficiency of microfinance institutions is significantly affected by total assets, cost per loan, loan per staff, legal status, and the county’s income group of the microfinance. Notably, the profit efficiency of microfinance institutions is adversely affected by the presence of female borrowers and female loan officers suggests that gender diversity plays a role in efficiency of microfinance institutions. Finally, we recommend that the managing body of microfinance work more on improving labor efficiency, earning asset utilization, loan collection efficiency, women’s involvement and the hottest technology implementation.

The manuscript is on an interesting topic. The findings are insightful. However, its current form cannot be published until and unless the concerns below are addressed:

ABSTRACT: In the abstract of the manuscript authors have made this recommendation: “Finally, we recommend that the managing body of microfinance work more on improving labor efficiency, earning asset utilization, loan collection efficiency, women’s involvement and the hottest technology implementation.” Please, on what findings have this recommendation been made? For instance, on what basis are authors have authors recommended more women involvement when they have stated in the second paragraph of the “Conclusions and Managerial Implications section” of their manuscript that “regarding gender diversity, the profit efficiency of microfinance institutions is adversely affected by the presence of female borrowers and female loan officers”? The recommendations of a study must be based on its findings. They should not spring from conjectures and imaginations!

Indeed, authors must rewrite the entire abstract to make it more concise devoid of unnecessary embellishment!

INTRODUCTION: Authors have not made a good case for their study. What gap(s) have authors attempted to fill or bridge? Accordingly, they are invited to improve the introduction to the manuscript by making a cogent case for it.

CONTRIBUTION OF THE STUDY: Contribution of the study should be rewritten. Authors must demonstrate the relevance of their study to the extant literature in terms of knowledge, managerial practice and theory. Authors have attempted to identify the contributions of their manuscript but what they have done is not convincing at all. Besides, authors have made inconsistent claims in the number of ways in which their study expands the frontiers of the existing knowledge.

“This study contributes to the existing stock of knowledge in three main ways” This appears in the last but one paragraph of the introduction.

“Moreover, the study findings contribute to the literature in two other ways.” This appears in the last paragraph of the theoretical and empirical review section of the paper.

Authors should clarify the number of ways the study in which their study contributes to the extant literature.

LITERATURE REVIEW: Authors have a dedicated section in their paper captioned “Theoretical and Empirical Review”. Under this section they have reviewed three theories: resource dependence theory, agency theory and social identity theory. However, they have failed to use these theories to explain or predict the relationship between the variables of interest in their study: Gender diversity and profit efficiency. This is fatal.

The empirical review part is too terse. Authors must improve it to make it possible for the reader to locate their findings in the empirical literature.

CONCLUSIONS AND MANAGERIAL IMPLICATIONS: This side of the manuscript requires a major improvement. What authors have written is nothing more than the rehash of the findings of the study. Besides, some vague statements have been made by authors. For example, authors have stated: “A uniform profit efficiency level was recognized by the MFIs in SSA from time to time due to the absence of self-advancement and management in line with the rapid changes in technology and technique.” What is the meaning of the above statement? Authors are advised to spend some time to draw inferences from their results as well as delineate the managerial implications of their findings.

LANGUAGE: “have been presented their annual financial report to the global microfinance information exchange” This should be corrected. I think “been” is not needed in the above phrase which appears in the first sentence under Sampling and Data source sub-section of the Material and Methods Section of the manuscript. Authors are advised to stick with one tense. They have mixed tenses. They should either write in the simple present tense or past tense. They should not mix tenses.

RECOMMENDATION: The manuscript is on an interesting topic that has the potential to make a valuable contribution to the extant literature. However, its current form cannot be published. It requires some significant improvement to bring it to the level suitable for publication. I have pointed out my main issues of concern in this report. Consequently, I recommend a major revision.

Reviewer #2: First and foremost, I would like to thank the Editor for giving me a chance to review this interesting paper.

This paper is well-structured and mainly focuses on the relationship between gender diversity and profit efficiency of MFIs in Sub-Saharan Africa.

My comments are as follows:

1. First, in the introduction part, the authors need to highlight the contribution of the paper in an efficient way. Specifically, why SSA countries are worth investigating? Why does profit efficiency matter?

Even when there was no research using SSA sample, your article may have marginal/little contribution if you are unable to emphasize the necessity of the research.

2. The review of literature is thin, separated into two parts. In the first part, the authors focus deeply on the determinants of efficiency. In the second part, the theoretical gounds on gender diversity and efficiency are stated.

Nonetheless, I suggest that the authors spend considerable time on the second one. Specifically, rich literature highlights the role of females on board of directors, but it is not the case for female borrowers and female managers. The authors need to state a strong statement on why there is a link between female borrowers/female managers and profit efficiency.

3. The authors employ one-step estimation. This is a good technique.

You have three key variables:

(i) Boardg: Broad gender diversity.

(ii) Mgmtg: Management gender diversity.

(iii) Femaleb: Female borrowers.

In Table 4, it could be a typo when the author wrote “Femalebr” instead of “Femaleb”. Kindly check this. In addition, in Table 1, you need to list all variables used in the model (I find that lnFemalestaff and Femaleof are not in Table 1)

4. Importantly, the authors are kindly suggested to find relevant (and sound) theories to explain the negative association between female borrowers and loan officers and profit efficiency.

5. Next, you employed a cross-country sample. Then, some country-level traits may influence profit efficiency. Why don’t you include some country-level controls such as economic development and conditions? Income group is ok, but such variable is time-invariant.

(6) Can you do a two-step estimation as a robustness check for your finding?

7. The description of profit efficiency score needs to be added. Otherwise, readers may be confused about this score.

8. An Appendix or a table of sample distribution (by year, country) will provide more comprehensive information on your dataset.

6. PLOS authors have the option to publish the peer review history of their article (what does this mean?). If published, this will include your full peer review and any attached files.

Reviewer #1: No

Reviewer #2: **Yes: **Duc Nguyen NGUYEN

---

## [Author Response · Author response to Decision Letter 0]

12 Feb 2024

Date: February 07, 2024

To: Academic Editor and Reviewer of PLOS ONE

Subject: Reponses to reviewer #1 comments

Hello dear all, I thank you very much for your time, constructive comments and supports regarding our manuscript titled “Gender diversity and profit efficiency of MFIs: A Sub-Saharan Africa study.”

Manuscript code: PONE – D – 23 – 42063 – EMID: 452ead2fc03182eb

Comment 1: ABSTRACT

- “Indeed, authors must rewrite the entire abstract to make it more concise devoid of unnecessary embellishment.”

Response 1: following your informative comment regarding the abstract of our study, we have modified the recommendation section (heighted in yellow) of the abstract based on the direct findings of the study. Dear reviewer, the rewritten abstract is included in the manuscript as it is presented below. 

Abstract

Irrespective of the promising opportunity to improve profit efficiency by at least 73%, microfinance institutions operating in Sub-Saharan Africa are efficient only for 27%, far below the average value. The conclusion is drawn after analyzing the profit efficiency of the microfinance institutions using the stochastic frontier approach applied to data obtained from 128 microfinance institutions operating in 34 Sub-Saharan African countries. The study results suggest the existence of uniform profit efficiency experience from time to time due to the lack of sufficient effort exerted to improve efficiency concurrent with the changes in technology and techniques of financial services provision. Microfinance institutions operating in low-income countries and credit union form microfinance are economically more efficient than their counterparts. Furthermore, the profit efficiency of microfinance institutions is significantly affected by total assets, cost per loan, loan per staff, legal status, and the county’s income group of the microfinance. Notably, the profit efficiency of microfinance institutions is adversely affected by the presence of female borrowers and female loan officers suggests that gender diversity plays a role in efficiency of microfinance institutions. Finally, we recommend that the managing body of microfinance institutions to work more on improving labor efficiency, earning asset utilization, loan collection efficiency, and implement self-improvement strategies to enhance the profit efficiency of the institutions.

Comment 2: INTRODUCTION: 

- “Authors have not made a good case for their study. What gap(s) have authors attempted to fill or bridge? Accordingly, they are invited to improve the introduction to the manuscript by making a cogent case for it.”

Response 2: Dear reviewer, following your constructive comments, we have tried to show the gaps of exiting literature and the contribution of our study in introduction section of the revised manuscript. The following paragraph is taken from the introduction section to show the need for study and tangible problems of MFIs that attract attention of scholars like us. The highlighted statement is added to the paragraph to describe statement of the problem in better way.

…… The stakeholders also argue for dual missions of MFIs in their operations. In the social mission, MFIs are designed to provide cheaper financial services to low-income people, women, and those excluded from mainstream financial institutions. In the economic mission, they are expected to generate profit from their operations to maintain a sustainable and stable financial position [9, 18]. However, the efficiency of MFIs is a prerequisite to achieve the missions [1]. Improving efficiency by working on profit maximization is an expressway for MFIs to reach their goals. Hence, scholars working inside and outside Africa are urging MFIs operating on the continent to focus on improving their technical efficiency [13], labor productivity [1, 35], wise utilization of resources and increasing output [31]. These opinions imply the institutions are economically inefficient in their operations in general and profitability in particular. Likewise, [1] suggested that the efficiency of MFIs in Africa is less than 41% implying the presence of wide space for further improvement in efficiency, by approximately 60% on average. 

Comment 3: CONTRIBUTION OF THE STUDY 

- “Contribution of the study should be rewritten and authors should clarify the number of ways the study in which their study contributes to the extant literature.”

Response 3: Dear reviewer, thank you very much for your in-depth and insightful comments. Regarding the contribution of the study, we have tried to see the effect of gender diversity from five perspectives – board member, management member, loan officer, regular personnel and borrower, which were not addressed by the exiting literature. Previous empirical studies were give emphases for gender diversity aspects from two perspectives such as board member and borrower gender diversity. However, as reports of the World Bank show (MIX, 2021), women may engage on at least three additional roles in MFIs. Namely, management member, loan officer, regular personnel, thus, we could extend the existing knowledge by adding three roles of women in MFIs to our study design. 

Regarding the number of contributions, the inconsistency is now corrected in the last paragraph under theoretical and literature review section and the corrected paragraph (highlighted in yellow) are presented as follows. 

…. This study aims to assess the association between women’s roles and the profit efficiency of MFIs in SSA from five aspects of gender diversity. Moreover, the generalizability of the study findings is improved for taking a large number of MFIs operating in all SSA countries for a longer period (2009 – 2018), and for applying a parametric panel data analysis technique, i.e., the stochastic frontier approach (SFA), unlike those studies adopted nonparametric data analysis approaches in the MFI literature. 

 Comment 4: LITERATURE REVIEW 

- “However, they have failed to use these theories to explain or predict the relationship between the variables of interest in their study: Gender diversity and profit efficiency.”

Response 4: Dear reviewer, we have tried to bridge the postulate of existing theories with the view of our study findings using the statements presented in the following paragraph (highlighted in yellow). The statement exists in the revised manuscript. We hope these statements well explained the relationship between gender diversity and profit efficiency. 

… Theoretically, these empirical findings present notable remarks. The adverse effects of borrowers’ and loan officers’ gender diversity against the profit efficiency of MFIs contradict the concepts of resource dependency theory, resource-based view and agency theory. However, the findings are consistent with the concept of social identity theory. It seems that the institutions serve women in borrowing services and allow them to hold officer positions, probably for the desire to meet regulation requirements for gender diversity at work. As a result, women receiving borrowing services from the institutions and working in the institutions may face group isolation and ignorance by their counterparts, and the institutions are limited in effectively extracting and using the inherent potential of women in the workplace.

Comment 5: CONCLUSIONS AND MANAGERIAL IMPLICATIONS

- “This side of the manuscript requires a major improvement. What authors have written is nothing more than the rehash of the findings of the study. Besides, some vague statements have been made by authors. For example, authors have stated: “A uniform profit efficiency level was recognized by the MFIs in SSA from time to time due to the absence of self-advancement and management in line with the rapid changes in technology and technique.” What is the meaning of the above statement? Authors are advised to spend some time to draw inferences from their results as well as delineate the managerial implications of their findings.”

Response 5: Following your constructive comment in this section, we have restated the previous conclusions and managerial implications in more informative ways as presented in the following passage (highlighted in yellow) and the main manuscript is rewritten accordingly. 

… MFIs operating in SSA realize profit efficiency below the average, only 27%. Credit union MFIs and MFIs operating in low-income countries are more efficient in realizing profit than other forms of MFIs and institutions operating in the upper-middle-income group. A uniform profit efficiency level was recognized by the MFIs over the study period that shows the absence of self-improvement in operations, interest expense management, human resource management and efficient utilization of earning assets in line with the rapid changes in financial services provision technology and techniques. Input prices such as the labor price and price of a fund and the output item, particularly other earning assets, play significant roles in the profit efficiency of MFIs operating in SSA.

Moreover, firm size, cost per loan, loan per personnel, legal status, and country’s income group significantly affect the profit efficiency of MFIs in SSA. Regarding gender diversity, the profit efficiency of microfinance institutions is adversely affected by the presence of female borrowers and female loan officers. It is necessary to work more on improving profit efficiency by designing alternative policy and updating existing strategies that could enhance labor productivity, reduce loan-related costs, capacitate women borrowers and increase loan collection efficiency.

To improve profit efficiency, MFIs in SAA are expected to undertake the following managerial actions and decisions. First, capacity-building reforms and training should be implemented to enhance labor productivity, ensure effective utilization of assets and reduce loan-operating costs. Second, redesign human resources policy and loan provision regulations to ensure women’s involvement on the board member, management, personnel and borrowing services that enable the MFIs to take gender diversity advantages in making sound decisions, resource provisions and cost management. Third, focus on removing work environment practices that expose women to self-categorization and isolation, and deteriorating their confidence to take full responsibility in decision-making. Generally, in the desire to improve profit efficiency, MFIs in SSA should reduce holding of the less productive labor force, minimize costly practices of producing loans, and make women empowerment and capacity building the core part of financial services provision. 

Comment 6: LANGUAGE

- “…have been presented their annual financial report to the global microfinance information exchange” This should be corrected. I think “been” is not needed in the above phrase which appears in the first sentence under Sampling and Data source sub-section of the Material and Methods Section of the manuscript. Authors are advised to stick with one tense. They have mixed tenses. They should either write in the simple present tense or past tense. They should not mix tenses.”

Response 6: Dear reviewer, the statement is corrected by removing “been” as … “have presented their annual financial report to the global microfinance information exchange …” in the revised manuscript. 

****** Thank you very much*******

Kind regards 

Tarekegn Tariku Ebissa

The corresponding author

---

## [Decision Letter · Decision Letter 1]

26 Jun 2024

PONE-D-23-42063R1Gender diversity and profit efficiency of microfinance institutions: A Sub-Saharan studyPLOS ONE

Dear Dr. Ebissa,

Thank you for submitting your manuscript to PLOS ONE. After careful consideration, we feel that it has merit but does not fully meet PLOS ONE’s publication criteria as it currently stands. Therefore, we invite you to submit a revised version of the manuscript that addresses the points raised during the review process.

**ACADEMIC EDITOR: ****Minor revision is required as follow:**1.
Revision requirements - Please address all reviewers comments satisfactorily. Highlight all changes in the text and provide details in the table of correction with indication of pages numbers for reviewer quick cross-check.2.
As indicated by the reviewer, we request the author to engage with a professional English editor to proofread the article. Please provide us the anonymous certificate of proofreading and attached it in the table of correction.==============================

We look forward to receiving your revised manuscript.

Kind regards,

Jasman Tuyon, Ph.D., MBA

Academic Editor

PLOS ONE

Journal Requirements:

Additional Editor Comments:

1. Revision requirements - Please address all reviewers comments satisfactorily. Highlight all changes in the text and provide details in the table of correction with indication of pages numbers for reviewer quick cross-check.

2. As indicated by the reviewer, we request the author to engage with a professional English editor to proofread the article. Please provide us the anonymous certificate of proofreading and attached it in the table of correction.

Reviewers' comments:

Reviewer's Responses to Questions

**Comments to the Author**

1. If the authors have adequately addressed your comments raised in a previous round of review and you feel that this manuscript is now acceptable for publication, you may indicate that here to bypass the “Comments to the Author” section, enter your conflict of interest statement in the “Confidential to Editor” section, and submit your "Accept" recommendation.

Reviewer #3: (No Response)

Reviewer #4: All comments have been addressed

2. Is the manuscript technically sound, and do the data support the conclusions?

Reviewer #3: Partly

Reviewer #4: Yes

3. Has the statistical analysis been performed appropriately and rigorously? 

Reviewer #3: Yes

Reviewer #4: Yes

4. Have the authors made all data underlying the findings in their manuscript fully available?

Reviewer #3: (No Response)

Reviewer #4: Yes

5. Is the manuscript presented in an intelligible fashion and written in standard English?

Reviewer #3: No

Reviewer #4: Yes

6. Review Comments to the Author

Reviewer #3: "The abstract is too revealing. It should be as concise as possible.

The introduction can be better written.

The first statement of the introduction states, “The stakeholders stated that microfinance institutions (MFIs) are vibrant tools for fighting poverty by reaching low-income people and disadvantaged groups such as women by creating affordable and easily accessible financial services” However, it is unclear which stakeholders this statement refers to, as a prior sentence seems to be missing. The same issue appears to be present in the second paragraph as well. The motivation for the study is not clear.

The establishment of the gap can be heightened.

There are a lot of recent studies on the topic such as :

Ebissa, T. T., Asfaw, A. S., & Lakew, D. M. (2024). Women’s participation and cost efficiency in microfinance institutions: a Sub-Saharan study. Cogent Business & Management, 11(1), 2304307.

Boadi, I., Dziwornu, R., & Osarfo, D. (2022). Technical efficiency in the Ghanaian banking sector: does boardroom gender diversity matter?. Corporate Governance: The International Journal of Business in Society, 22(5), 1133-1157.

Sarpong‐Danquah, B., Adusei, M., & Magnus Frimpong, J. (2023). Effect of board gender diversity on the financial performance of microfinance institutions: Does judicial efficiency matter?. Annals of Public and Cooperative Economics, 94(2), 495-518.

Adalessossi, K. (2024). What are the determinants of the financial and social performance of MFIs in Togo? Does gender borrower matter on financial performance?. Finance Research Letters, 105192.

Ali, H., Gueyie, J. P., & Chrysostome, E. V. (2023). Gender, credit risk and performance in sub-Saharan African microfinance institutions. Journal of African Business, 24(2), 235-259.

Bouslah, K., Li, Q., & Mobarek, A. (2022). Board gender diversity and the social performance of microfinance institutions. Working Papers In Responsible Banking & Finance, 23(11), 1-55.

Díaz‐Martín, S., Feria‐Dominguez, J. M., & Naranjo‐Gil, D. (2022). Are microfinance institutions' financial performance gender driven? Evidence from Argentina. Business Strategy & Development, 5(3), 197-208.

What makes your work different from the above listed studies? It is important to cite these studies and indicate how they differ from your study.

It is important for authors to make sure that their introduction is interesting and engaging to capture the reader's attention.

In the empirical literature review, the candidate should critique the existing literature rather than merely stating them.

It is advisable that authors cite recent studies(2024-2022) such as : Ebissa, T. T., Asfaw, A. S., & Lakew, D. M. (2024; MI Hossain, MA Mia, L Dalla Pellegrina , 2024; Ali, H., Gueyie, J. P., & Chrysostome, E. V. (2023; Sarpong‐Danquah, B., Adusei, M., & Magnus Frimpong, J. (2023; Díaz‐Martín, S., Feria‐Dominguez, J. M., & Naranjo‐Gil, D. (2022). Etc.

It appears the conclusion of the study in virtually restating the findings of the study. The authors should give an overview of the study while helping readers to reflect on what they just read, draw connections to existing knowledge, and spark their desire to further explore the subject.

Language should improve massively.

Reviewer #4: I am satisfied with all the amendments done in the paper. The authors have undertaken the required revision as per the comments raised earlier.

7. PLOS authors have the option to publish the peer review history of their article (what does this mean?). If published, this will include your full peer review and any attached files.

Reviewer #3: No

Reviewer #4: No

---

## [Author Response · Author response to Decision Letter 1]

29 Jun 2024

Subject: Reponses to reviewer #3 and editor comments

Hello dear all, I thank you very much for your time, constructive comments and supports regarding our manuscript titled “Gender diversity and profit efficiency of MFIs: A Sub-Saharan Africa study.”

Revision code: PONE-D-23-42063R1

Manuscript code: PONE – D – 23 – 42063 – EMID: 452ead2fc03182eb

Reviewer #3 comments

1. Abstract 

- "The abstract is too revealing. It should be as concise as possible”

Authors’ response

Dear reviewer, earlier abstract is revised following your constructive comments by removing long statement and paraphrasing sentences. The revised version of the abstract is presented below and changes are highlighted in yellow. 

“Irrespective of the promising opportunity to improve profit efficiency by at least 73%, microfinance institutions operating in Sub-Saharan Africa are efficient only for 27%, far below the average value. The conclusion is drawn after analyzing the profit efficiency of the microfinance institutions using the stochastic frontier approach applied to data obtained from 128 microfinance institutions operating in 34 Sub-Saharan African countries. The study results suggest the presence of uniform profit efficiency experience across time among microfinance institutions. Microfinance institutions operating in low-income countries and credit union form microfinance are economically more efficient than their counterparts. Furthermore, the profit efficiency of microfinance institutions is significantly affected by total assets, cost per loan, loan per staff, legal status, and the county’s income group of microfinance. Notably, the profit efficiency of microfinance institutions is adversely affected by the presence of female borrowers and female loan officers suggesting that gender diversity plays a role in the efficiency of microfinance institutions. Finally, we recommend that the managing body of microfinance work more on improving labor efficiency, earning asset utilization, loan collection efficiency, women’s involvement and the hottest technology implementation.”

2. Introduction

“The introduction can be better written”

Authors’ response

Dear reviewer, the introduction sections are rewritten by removing the some confusing phrases and statement. The revised introduction sections for paragraph one and two are presented below with heighted mark and included in the revised manuscript as well.

“Microfinance institutions (MFIs) are vibrant tools for fighting poverty by reaching low-income people and disadvantaged groups such as women by creating affordable and easily accessible financial services. In addition, MFIs play remarkable roles in economic growth by providing services to rural people, reducing transaction costs, easing loan requirements (collateral), smoothing consumption, ensuring gender equality, and lending to small-scale borrowers [4, 9, 24, 29]. The roles are more helpful and fruitful in the SSA region than in other parts of the world since the region is occupied by the least banked households and the highest number of low-income people [19]. The good news is that approximately 917 MFIs are currently operating in the SSA region [26].

Furthermore, MFIs have dual missions to achieve in their operations. In the social mission, MFIs are designed to provide cheaper financial services to low-income people, women, and those excluded from mainstream financial institutions. In the economic mission, they are expected to generate profit from their operations to maintain a sustainable and stable financial position [9, 18]. However, the efficiency of MFIs is a prerequisite to achieving these missions [1]. Improving efficiency by working on profit maximization is an expressway for MFIs to reach their goals. Hence, scholars working inside and outside Africa are urging MFIs operating on the continent to focus on improving their technical efficiency [13], labor productivity [1, 35], wise utilization of resources and increasing output [31].”

3. Motivation of the study and research gap

Authors’ response

Dear reviewer, in this revision we have presented additional empirical literature gaps by critiquing the limitations of recent studies and showing the contributions (motivations) of our study. The revised section is presented below and the changes are highlighted. 

-------- Despite the above facts, very few studies have been conducted to investigate the profit efficiency of MFIs operating in SSA, and the empirical gaps are remarkable. Empirical studies in the literature are limited in applying stochastic frontier analysis (SFA) – the parametric approach – [3, 9, 31], covering a large number of MFIs [29] and assessing the roles of gender diversity in the profit efficiency of MFIs. For instance, a recent study assessed the effect of women’s participation on the cost efficiency of MFIs in Sub-Sahara Africa fails to address the profit efficiency aspects of the institutions [40]. Unlike our study, the empirical works of [41, 42, 43, 44] focused on addressing the effects of gender diversity on financial performance, social performance and technical efficiency of MFIs in Sub-Saharan Africa. Rather our study emphasized the effects of gender diversity on the profit efficiency of the institutions. 

4. Conclusions of the study 

Authors’ response 

Dear reviewer, we tried to draw the conclusions of our study from the findings of the study and we could have a meaningful recommendation and managerial implications for MFIs operating in the study area. The revised version of conclusion section is presented below with highlighted marks. 

“MFIs operating in SSA realize profit efficiency below the average, only 27%. It indicates the presence of potential opportunities to raise the current profitability position of MFIs in SSA by improving their efficiency level by 73%. Credit union MFIs and MFIs operating in low-income countries are more efficient in realizing profit efficiency than other forms of MFIs and institutions operating in the upper-middle-income group. A uniform profit efficiency level was recognized by the MFIs in SSA from time to time due to the absence of self-advancement and management in line with the rapid changes in technology, and financial services provision techniques. Input prices such as the labor price and price of a fund and the output item, particularly other earning assets, play significant roles in the profit efficiency of MFIs operating in SSA.”

Response to editor’s comment and journal requirement 

Dear editor, we have cited five new empirical studies in the body of the text and included in the reference section in response to the reviewer comments to revise empirical research gaps and clarify motivation of our study. In addition, the presence of all in text citation and inclusion in the reference are verified rigorously. 

Dear editor and reviewer, we thank you very much. 

Kind regards 

Tarekegn Tariku Ebissa

The corresponding author 

Note: table of correction is attached with separate response letter for your quick crosscheck.

---

## [Decision Letter · Decision Letter 2]

11 Jul 2024

Gender diversity and profit efficiency of microfinance institutions: A Sub-Saharan study

PONE-D-23-42063R2

Dear Dr. Ebissa,

We’re pleased to inform you that your manuscript has been judged scientifically suitable for publication and will be formally accepted for publication once it meets all outstanding technical requirements.

Kind regards,

Jasman Tuyon, Ph.D., MBA

Academic Editor

PLOS ONE

Additional Editor Comments (optional):

Accepted as per the reviewer reports

Reviewers' comments:

Reviewer's Responses to Questions

**Comments to the Author**

1. If the authors have adequately addressed your comments raised in a previous round of review and you feel that this manuscript is now acceptable for publication, you may indicate that here to bypass the “Comments to the Author” section, enter your conflict of interest statement in the “Confidential to Editor” section, and submit your "Accept" recommendation.

Reviewer #3: All comments have been addressed

2. Is the manuscript technically sound, and do the data support the conclusions?

Reviewer #3: Yes

3. Has the statistical analysis been performed appropriately and rigorously? 

Reviewer #3: Yes

4. Have the authors made all data underlying the findings in their manuscript fully available?

Reviewer #3: (No Response)

5. Is the manuscript presented in an intelligible fashion and written in standard English?

Reviewer #3: Yes

6. Review Comments to the Author

Reviewer #3: All the raised issues have been addressed. The authors removed some parts of the abstract. The introduction is better written and free from the previously identified errors. The authors made efforts to enhance the statement of the gap, and the conclusion is solid. Well done!

7. PLOS authors have the option to publish the peer review history of their article (what does this mean?). If published, this will include your full peer review and any attached files.

Reviewer #3: No

---

## [Editor Report · Acceptance letter]

26 Jul 2024

PONE-D-23-42063R2 

PLOS ONE

Dear Dr. Ebissa, 

I'm pleased to inform you that your manuscript has been deemed suitable for publication in PLOS ONE. Congratulations! Your manuscript is now being handed over to our production team.

Kind regards, 

on behalf of

Dr. Jasman Tuyon 

Academic Editor

PLOS ONE